# Effects of the Overexpression of Progesterone Receptors on a Precancer p53 and Rb-Defective Human Fallopian Tube Epithelial Cell Line

**DOI:** 10.3390/ijms241411823

**Published:** 2023-07-23

**Authors:** Yu-Hsun Chang, Kun-Chi Wu, Kai-Hung Wang, Dah-Ching Ding

**Affiliations:** 1Department of Pediatrics, Hualien Tzu Chi Hospital, Buddhist Tzu Chi Medical Foundation, Tzu Chi University, Hualien 97004, Taiwan; cyh0515@gmail.com; 2Department of Orthopedics, Hualien Tzu Chi Hospital, Buddhist Tzu Chi Medical Foundation, Tzu Chi University, Hualien 97004, Taiwan; drwukunchi@yahoo.com.tw; 3Department of Medical Research, Hualien Tzu Chi Hospital, Buddhist Tzu Chi Medical Foundation, Tzu Chi University, Hualien 97004, Taiwan; kennyhug0201@gmail.com; 4Department of Obstetrics and Gynecology, Hualien Tzu Chi Hospital, Buddhist Tzu Chi Medical Foundation, Tzu Chi University, Hualien 97004, Taiwan; 5Institute of Medical Sciences, Tzu Chi University, Hualien 97004, Taiwan

**Keywords:** progesterone receptors, FE25, fallopian tube epithelial cells, p53, retinoblastoma

## Abstract

This study investigated the effects of progesterone receptors A (PRA) and B (PRB) on proliferation, migration, invasion, anchorage-independent growth (AIG), and apoptosis of FE25 cells, a precancer p53- and retinoblastoma-defective human fallopian tube epithelial cell line. We observed that the transfection of PRA (FE25-PRA) or PRB (FE25-PRB) into FE25 cells significantly increased the expression of PRA or PRB at both RNA and protein levels without affecting cell morphology. The FE25-PRA cells exhibited slower proliferation, whereas FE25-PRB showed faster cell proliferation than the control cells. In contrast, the FE25-PRA cells showed the highest migration and invasion abilities, whereas the FE25-PRB cells showed the lowest migration and invasion abilities. After treatment with progesterone, all cell types showed decreased AIG levels, increased apoptotic rates in Terminal deoxynucleotidyl transferase (TdT) dUTP nick end labeling assay (TUNEL) staining, and increased levels of apoptotic proteins ascertained based on cleaved caspase-3 levels. The half-maximal inhibitory concentration of carboplatin increased in FE25-PRB cells, but that of paclitaxel remained unchanged. Overall, this study suggests that PRA and PRB have distinct roles in regulating the behavior of FE25 cells, and targeting these receptors could be a potential therapeutic strategy for ovarian cancer treatment. If PRA or PRB overexpression is observed in high-grade serous carcinoma, progesterone could be considered as an adjuvant therapy for these specific cancer patients. However, further research is needed to confirm these findings and investigate the mechanisms underlying these effects.

## 1. Introduction

Epithelial ovarian cancer (OC) is the fifth leading cause of cancer-related deaths in women, with high-grade serous ovarian carcinoma being the most lethal [1,2]. A total of 313,959 new OC cases are recorded globally [2]. Most patients with OC are diagnosed with late-stage disease, and the 5-year survival is only 30–40% [3]. Current treatment options include debulking surgery and adjuvant chemotherapy.

Besides the above therapy, targeted therapies utilizing anti-vascular endothelial growth factor (VEGF) agents and poly ADP ribose polymerase (PARP) inhibitors have emerged as treatment options for OC [4]. These therapies have shown promising results in improving patient outcomes and are being incorporated into OC treatment strategies [4]. Response rates with bevacizumab-containing regimens range from approximately 30% to 60% in OC [5,6]. In clinical trials, PARP inhibitors have shown response rates ranging from approximately 40% to 60% in patients with BRCA-mutated OC [7,8].

Despite the above treatments, progesterone treatment when the progesterone receptor (PR) is present may be another treatment option for OC [9]. Previous studies have reported a variable response rate of antiprogesterone therapy, ranging from 5% to 34% [10]. In a retrospective cohort study, it was found that strong expression of progesterone receptor B (PRB) was associated with improved platinum sensitivity and overall survival [9]. Additionally, there was a suggestion that progesterone might enhance the effectiveness of platinum-based chemotherapy, indicating its potential as a platinum sensitizer [9]. Hence, PR status not only serves as a prognostic factor but also as a marker for chemotherapy efficacy in OC. Moreover, PRs have been shown to have distinct roles in regulating the behavior of OC. Understanding these roles is crucial in comprehending the mechanisms underlying the disease. Furthermore, it is important to consider the economic burden associated with targeted therapies, which tends to be significantly higher compared to hormone therapies. Consequently, exploring the involvement of PRs in OC holds considerable significance in terms of both clinical outcomes and healthcare costs.

Progesterone receptors A (PRA) and B (PRB) are two isoforms of the progesterone receptor expressed from a single gene through alternative splicing [11]. The PRA lacks a 164-amino-acid segment in its N-terminus, resulting in a shorter protein with a weaker transcriptional activation function than the PRB. The PRB contains this 164-amino-acid segment, which allows for a more robust transcriptional activation function and confers unique activities, such as ligand-independent activation and regulation of PRA activity [12]. Several studies have investigated the differential roles of PRA and PRB in various cellular processes, including cell proliferation, differentiation, and migration [11]. The PRA and PRB have distinct and opposing roles in OC, and their expression levels may affect patient outcomes and responses to therapy.

The expression of PRs is also a prognostic factor of OC [13]. Patients with OC who were estrogen receptor (ER)- and PR-positive had better outcomes [14]. Loss of PR and ER also correlated with an increase in OC grading [15]. In a previous study, the tumors that were PR-negative were associated with poor survival [16]. Another study that recruited 2933 patients with OC showed that PR expression was associated with favorable survival in patients with endometrioid and high-grade serous carcinoma [17]. In addition, PRs are either predominantly downregulated, or their expression is lost in OCs [18]. Two progesterone receptor gene (*PGR*) polymorphisms are associated with an increased risk of OC [19]. The PRs play an essential role in protection against OC, and the downregulation of PRs is a prerequisite for carcinogenesis.

OC is divided into several subtypes: serous, mucinous, endometrioid, clear cell, Brenner tumor, and undifferentiated carcinoma [20]. The precursor lesion of serous carcinoma is the fallopian tube epithelium (FTE). Mucinous carcinoma may have originated from germ cells. Endometrioid and clear cell carcinoma may have originated from endometrial tissue. Brenner tumors may be originated from transitional cells [20].

Type 2 OC (high-grade serous carcinoma, HGSC) originates from FTE [21]. We previously derived an FTE cell line (FE25) that exhibited precancerous characteristics due to p53 and retinoblastoma (Rb) deletions [22]. Highly passaged FE25 cells also exhibit type 2 OC characteristics and might be suitable for OC research [23]. However, the effects of PR on FE25 cells are still unknown [23].

This study aimed to explore the roles of PRA and PRB on the proliferation, migration, invasion, anchorage-independent growth (AIG), half-maximal inhibitory concentration (IC50) of chemotherapy drugs, and apoptosis of FE25 cells after treatment with progesterone (P4).

## 2. Results

### 2.1. Transfection with PRA- or PRB-Overexpressing Constructs Increased the Expression of PRA or PRB in FE25 Cells

After the transfection of PRA- and PRB-overexpressing constructs into FE25 cells, the morphology of cells did not change and revealed a cuboidal shape, which was characteristic of epithelial cells (Figure 1A). qPCR was performed to determine the RNA expression levels of PRA and PRB after transfection. Both PRA and PRB were significantly overexpressed in PRB-transfected FE25 cells compared to that in untransfected FE25 cells (*p* < 0.001, Figure 1B). PRA-transfected FE25 cells exhibited increased expression of PRA, while PRB expression remained unchanged compared to FE25 cells (Figure 1B). To validate these findings at the protein level, we performed Western blotting to assess the levels of PRA and PRB proteins following transfection. Overexpressed PRA was noted in PRA-transfected FE25 cells (Figure 1C). Overexpression of both PRA and PRB proteins in PRB-transfected FE25 cells was found compared to untransfected or control vector-transfected FE25 cells (Figure 1C). Notably, overexpression of PRB led to an increase in both PRB mRNA and protein expression, as illustrated in Figure 1B,C. Additionally, we examined PRB protein expression at passages 3 and 14 and found consistent and stable expression levels (Figure 1C). Collectively, the morphology of FE25 cells remained unaltered following PRA or PRB overexpression. PRA overexpression resulted in increased PRA mRNA and protein expression, while PRB overexpression led to increased PRB mRNA and protein expression.

### 2.2. FE25-PRA Decreased Proliferation, While FE25-PRB Increased Proliferation

FE25-PRA cells’ proliferation rate was significantly lower than that of untransfected FE25 and FE25-lenti-ctrl cells (Figure 2A). Nevertheless, in FE25-PRB cells, the cell proliferation rate was significantly higher than that of untransfected FE25 and FE25-lenti-ctrl cells (Figure 2B). In summary, PRA overexpression would decrease cell proliferation, and PRB overexpression would increase cell proliferation.

### 2.3. FE25-PRA Promoted Cell Migration, While FE25-PRB Inhibited Cell Migration, Both of Which Were Reversed by Progesterone Treatment

The migration rate of FE25-PRA was higher than that of the other cells (Figure 3A,B). In contrast, the migration rate of FE25-PRB was lower than that of the other cells (Figure 3A,B). However, after P4 treatment, cell migration decreased in FE25 and FE25-lenti-ctrl cells. In FE25-PRA cells, the migration did not significantly decrease after adding P4. In FE25-PRB cells treated with P4, the migration was increased significantly than the control (Figure 3B). In summary, PRA overexpression enhanced migration, but PRB overexpression decreased migration. After adding progesterone, PRB overexpression could enhance migration.

### 2.4. FE25-PRA Promoted Cell Invasion, While FE25-PRB Inhibited Cell Invasion, Both of Which Were Reversed by Progesterone Treatment

Cell invasion rate was the fastest in FE25-PRA (Figure 4A,B). In contrast, the cell invasion rate was the lowest in FE25-PRB (Figure 4A,B). However, after P4 treatment, cell invasion decreased, except in the FE25-PRB cells (Figure 4). FE25-PRB showed a decreased invasion capability than the other three cell lines. After adding progesterone, FE25-PRB increased the invasion more than the other three cell lines (Figure 4). In summary, PRA overexpression enhanced invasion but PRB overexpression decreased invasion. After adding progesterone, PRB overexpression could enhance invasion.

### 2.5. Progesterone Inhibited AIG

To evaluate FE25 proliferation in soft agar with or without overexpression of PRA or PRB, AIG for 2 weeks was carried out. The colony number of FE25 cells remained unchanged after transfection with PRA- or PRB-bearing constructs (Figure 5A–D). After progesterone treatment, colony numbers significantly decreased in all cell types (*p* < 0.001, Figure 5A–D). In summary, PRA or PRB did not change AIG capability but decreased AIG capability after adding progesterone in all four cell lines.

### 2.6. Progesterone Increased Apoptosis (TUNEL^+^ Cells) in FE25-PRA or FE25-PRB

Apoptosis in FE25 cells did not increase after transfection with PRA- or PRB-bearing constructs. After treatment with progesterone, all cell types showed an increase in TUNEL-positive cells (Figure 6A–D). The FE25 and FE25-lenti-ctrl showed 8% of apoptotic cells after progesterone treatment (Figure 6A,B). The FE25 cells transfected with PRA- or PRB-bearing constructs showed 30–40% TUNEL-positive cells (Figure 6C,D). In summary, after PRA or PRB overexpression, cell apoptosis would not increase. After adding progesterone, PRA or PRB overexpression cells showed more apoptosis than FE25 or FE25-lenti-ctrl cells.

### 2.7. FE25-PRA or -PRB Increased Chemoresistance of Carboplatin

The IC50 values of carboplatin and paclitaxel were evaluated in FE25 cells overexpressing PRA or PRB. The IC50 of carboplatin was increased in FE25-PRA or FE25-PRB cells compared to that in FE25-lenti-ctrl cells (Figure 7A–C). The IC50 value of paclitaxel was the same in the three cell types (Figure 7D–F). In summary, overexpression of PRB resulted in an increased IC50 of carboplatin. However, there was no observed change in the IC50 values of paclitaxel following overexpression of PRA or PRB.

### 2.8. Progesterone Activated the AKT and ERK Signaling Pathways in FE25-PRA or FE25-PRB

PRA or PRB expression has been shown to elevate AKT levels potentially, and treatment with P4 (progesterone) leads to an increase in phosphorylated AKT (p-AKT). Additionally, PRA or PRB expression may increase AKT and ERK levels, although it does decrease phosphorylated ERK (p-ERK) levels. The inhibitory effect on pERK levels caused by PRB expression can be reversed with P4 treatment, as depicted in Figure 8. In summary, progesterone could potentially enhance the phosphorylation of AKT and ERK signaling pathways in FE25-PRA and FE25-PRB cells.

### 2.9. Progesterone Decreased BCL2 and XIAP Expression in FE25-PRA or FE25-PRB

Treatment with 100 μM progesterone resulted in a reduction of BCL2 and XIAP protein expression in FE25, FE25-lenti-ctrl, and FE25-PRA cells, as illustrated in Figure 9. However, in FE25-PRB cells, progesterone treatment did not decrease the levels of BCL2 or XIAP expression, as shown in Figure 9. The results indicate that treatment with 100 μM progesterone effectively decreased the expression of BCL2 and XIAP proteins in FE25, FE25-lenti-ctrl, and FE25-PRA cells. However, in FE25-PRB cells, progesterone treatment failed to reduce the levels of BCL2 or XIAP expression. These results suggest that the presence of PRB may influence the response to progesterone treatment in terms of BCL2 and XIAP expression.

### 2.10. Progesterone Treatment Increased Cleaved Caspase 3 Expression in FE25-PRB but Decreased in FE25-PRA

Treatment of FE25, FE25-lenti-ctrl, and FE25-PRB cells with 100 μM progesterone resulted in an increase in cleaved caspase-3 expression, as depicted in Figure 10. However, progesterone treatment led to a decrease in cleaved caspase-3 expression in FE25-PRA cells (Figure 10). In summary, progesterone was found to increase cleaved caspase-3 expression in FE25, FE25-lenti-ctrl, and FE25-PRB cells, while it had the opposite effect in FE25-PRA cells. Progesterone was found to increase cleaved caspase-3 expression in FE25, FE25-lenti-ctrl, and FE25-PRB cells, indicating its pro-apoptotic effect in these cell types. However, in FE25-PRA cells, progesterone had the opposite effect, suggesting a distinct response in terms of cleaved caspase-3 expression. Overall, these findings highlight the complex and context-dependent effects of progesterone on cellular pathways and apoptotic processes in different cell lines.

## 3. Discussion

Our observations revealed that transfecting PRA- or PRB-overexpressing constructs into FE25 cells significantly increased the expression of PRA or PRB at both RNA and protein levels without affecting cell morphology. PRA-transfected cells exhibited slower proliferation, while PRB-transfected cells showed faster proliferation than control cells. Before progesterone treatment, FE25-PRA cells displayed enhanced migration and invasion abilities compared to FE25 and FE25-lenti-ctrl cells, whereas FE25-PRB cells exhibited reduced migration and invasion abilities compared to FE25 and FE25-lenti-ctrl cells. Following progesterone treatment, migration, and invasion abilities were increased in FE25-PRA and FE25-PRB cells compared to FE25 and FE25-lenti-ctrl cells. Moreover, after progesterone treatment, all cell types demonstrated decreased AIG levels, increased apoptotic rates, and elevated levels of apoptotic proteins. However, the IC50 levels of carboplatin were increased in FE25 cells transfected with PRA- or PRB-overexpressing constructs, while the IC50 levels of paclitaxel remained unchanged. In PRB-overexpressing FE25 cells, progesterone has the potential to enhance the phosphorylation of AKT and ERK signaling pathways. Additionally, progesterone was found to increase cleaved caspase-3 expression in FE25, FE25-lenti-ctrl, and FE25-PRB cells, indicating its pro-apoptotic effect in these cell types. However, in FE25-PRA cells, progesterone had the opposite effect, suggesting a distinct response in terms of cleaved caspase-3 expression.

The observed phenomenon of increased protein and mRNA expressions of both PRA and B when PRB was overexpressed could be explained by several mechanisms. Positive feedback loop: PRA and B may act in a positive feedback loop, where the overexpression of PRB stimulates the expression of both PRA and B. This means that the increased levels of PRB could enhance the transcription and translation processes involved in producing both PRA and B [24]. Co-regulation: PRA and B genes may be co-regulated by common regulatory elements or factors. When PRB is overexpressed, it could lead to the activation or upregulation of these shared regulatory elements, resulting in increased expressions of both PRA and B [25]. Stabilization of mRNA transcripts: The overexpression of PRB could influence the stability and lifespan of mRNA transcripts for both PRA and B. The increased levels of PRB may enhance the stability of the mRNA molecules, leading to increased protein production of both PRA and B [26]. Cross-talk between signaling pathways: PRA and B may be involved in overlapping signaling pathways, where the overexpression of PRB could activate or modulate these pathways. This cross-talk between signaling pathways could result in the upregulation of both PRA and B at the transcriptional and translational levels [12].

The observed reversal in phenotype between FE25-PRB cells and FE25-PRA cells when treated with progesterone suggests that the effects of progesterone on these cells are mediated through differential activation of progesterone receptor isoforms. Progesterone receptor isoforms, PRB and PRA, are two major variants of the progesterone receptor that can have distinct functional properties. PRB is known to have a higher DNA-binding affinity and transcriptional activity compared to PRA [27]. It is associated with the activation of genes involved in differentiation and inhibition of cell proliferation, while PRA is generally associated with weaker transcriptional activity. Without progesterone treatment, FE25-PRB cells exhibit lower migratory and invasive capacities than FE25-PRA cells. This could be due to the differential effects of PRB and PRA on the expression of genes involved in cell migration and invasion [11]. The higher transcriptional activity of PRB may result in the upregulation of genes that inhibit migration and invasion, leading to the observed phenotype. However, when progesterone is introduced, it can bind to both PRB and PRA isoforms, activating their transcriptional activity. Progesterone-bound PRA may enhance the expression of genes associated with migration and invasion, while progesterone-bound PRB may inhibit the expression of these genes. This shift in gene expression profiles could potentially reverse the migratory and invasive capacities of FE25-PRB cells, making them more similar to FE25-PRA cells. In summary, the phenotype reversal observed after progesterone treatment suggests that progesterone modulates the differential effects of PRB and PRA isoforms on gene expression, leading to a switch in the migratory and invasive capacities of the cells.

PRA and PRB, are the two isoforms of the PR protein produced from a single gene, *PGR*. The PRA is the shorter isoform of the progesterone receptor and has a dominant-negative effect on PRB activity. It is mainly expressed in the endometrium and is involved in regulating cell proliferation and differentiation, as well as in the immune response [28], and is required for the maintenance of pregnancy and prevention of preterm birth [29]. In contrast, PRB is the longer isoform of the progesterone receptor and is expressed in a wide range of tissues, including the breast, ovary, uterus, and brain [30]. It has more potent transcriptional activity than PRA and regulates the menstrual cycle, fertility, and pregnancy [29], and is also crucial for the development and differentiation of the mammary gland and in breast cancer progression [31]. A previous study reported that PRA and PRB play distinct roles in mammary gland development and that PRA is required for lobuloalveolar development. However, PRB is dispensable for this process [32]. Another study has suggested that PRA and PRB play different roles in regulating trophoblast invasion during pregnancy [33]. In our study, PRA and PRB exerted distinct effects on FE25 cell activity.

A previous study has examined the effects of progesterone treatment on different histological subtypes of ovarian carcinoma. They observed a significant reduction in the survival of patients with endometrioid ovarian carcinoma after progesterone treatment [34]. This effect was related to PR presence; a 43% reduction in cell number was observed in PR-positive endometrioid ovarian carcinoma [34]. Our previous study also showed that PR loss was an essential step in serous tubal intraepithelial carcinoma development, in which FE25 cells underwent necroptosis if PR was enhanced exogenously [35,36]. A previous study showed that PRB overexpression in OC cells, i.e., OC-3-VGH and OVCAR-3, increased cisplatin sensitivity by promoting apoptosis [9]. However, one study has shown that progesterone is critical for OC development [37]. Deletion of PR expression suppressed OC metastasis, enhanced sensitivity to chemotherapy drugs, and improved mouse survival [37]. In the current study, we overexpressed PRA and PRB in FE25 cells and observed increased apoptosis following progesterone treatment. PRB increased FE25 cell proliferation but decreased migration and invasion activities.

PR expression is frequently detected in OC and is associated with improved patient outcomes [38]. However, the differential expression of PR isoforms A and B can vary among ovarian tumors and different cell lines [39]. The PRA is the most abundant isoform and has been shown to promote cell proliferation and tumor growth in a few studies [40]. In contrast, PRB has been reported to have anti-proliferative effects and can inhibit the growth of OC cells [41]. Additionally, PRB has been associated with increased sensitivity to progestins such as medroxyprogesterone acetate (MPA), which is commonly used to treat OC [42]. PRA has a remarkable ability to activate gene expression in response to progestins, whereas PRB interacts with co-regulators to inhibit PRA activity [30]. In our study, FE25 cells did not initially express PRA or PRB. After PRA or PRB transfection, proliferation, migration, invasion, and apoptosis after progesterone treatment differed between PRA and PRB-FE25 cells.

Progesterone receptor isoforms, PRA and PRB, can potentially serve as biomarkers in clinical practice for OC. It can predict response to hormone therapy, provide prognostic indicators, differentiate subtypes of OC, and combine other biomarkers such as estrogen receptors. It can provide a more comprehensive molecular profile of the tumor and aid in personalized treatment strategies.

The limitation of this study was that the concentration of P4 used was higher than the physiological concentration. It might act on many receptors and so off-target effects would be possible.

## 4. Materials and Methods

### 4.1. Cell Lines

The FE25 cell line was employed. The FE25 cells were cultured in MCDB105 and M199 medium (M6395, M4530, Sigma-Aldrich, St. Louis, MO, USA) plus 10% fetal bovine serum (Biological Ind., Kibbutz, Israel), 100 IU/mL penicillin, and 100 μg/mL streptomycin (Sigma). The cells were cultivated in a 75 cm^2^ culture surface per culture flask in 12 mL of medium culture and were incubated at 37 °C temperature in an incubator containing 5% CO_2_.

### 4.2. Upregulation of Progesterone Receptors Using a Lentivector

PRA and PRB lentivectors (Human, CMV (cytomegalovirus), pLenti-GIII-CMV-GFP-2A-Puro) or empty control were purchased from ABM Company (Richmond, BC, Canada). Transfection was performed according to the manufacturer’s instructions. A multiplicity of infection of five viral particles per cell was used. The FE25 cells transfected with PRA and PRB constructs are referred to as FE25-PRA and FE25-PRB, respectively. Cells transfected with the empty control vector are denoted as FE25-lenti-ctrl. The FE25 cells were infected with lentivectors twice and selected in the presence of puromycin. The resulting cells were maintained under original cell culture conditions and were compared with the original FE25 cells.

### 4.3. The Proliferation of Cells

Cell proliferation was assessed using 2,3-bis(2-methoxy-4-nitro-5-sulfophenyl)-5-[(phenylamino) carbonyl]-2*H*-tetrazolium hydroxide (XTT) assay. The tested cells were plated in one well of a 96-well at a density of 2 × 10^3^ cells with 100 μL culture medium. The optical density of cells was measured on days 0, 3, 5, and 7. The tested cells were incubated with 150 μL XTT solution (Biological Industries, Beit-Haemek, Israel) for 3 h at 37 °C in an incubator. A microplate reader (Bio-Rad Model 3550, Hercules, CA, USA) was used to measure the optical density at 450 nm. The optical density values at each time point were used to construct proliferation curves of the tested cell lines.

### 4.4. Migration and Invasion Assay

Placed 5 × 10^4^ tested cells (FE25, FE25-lenti-ctrl, FE25-PRA, and FE25-PRB cells) with 200 μL medium into the upper layer of a 24-well transwell Boyden chamber (8 μm pore size; Costar, Corning Inc., Corning, NY, USA), and allowed the cells to migrate to the lower layer, where there were no cells but only 500 μL culture medium. After culturing for 48 h, the migrated cells were stained with crystal violet (Sigma-Aldrich), and the cells were counted with an optical microscope. All experiments were performed in triplicates.

Then, 5 × 10^4^ tested cells with 200 μL medium were placed into the upper chamber coated with Matrigel (8 μm pore size; Costar, Corning Inc.) and the cells were allowed to invade neighboring cells. After 48 h of culture, cells were stained with crystal violet (Sigma-Aldrich). The number of invading cells was counted. The experiments were repeated thrice.

### 4.5. Anchorage-Independent Growth in Soft Agar

Then, 4 mL culture medium was mixed with 0.35% agarose and 5 × 10^5^ tested cells (FE25, FE25-lenti-ctrl, FE25-PRA, and FE25-PRB cells), and 5 mL of a 0.7% agarose base was placed onto the lower layer. The experiment was repeated thrice. After 14 days of culture, the cells were stained with 0.8 mM crystal violet (Sigma-Aldrich), and the number of colonies was counted.

### 4.6. Quantitative Polymerase Chain Reaction (qPCR) and Reverse Transcription-Polymerase Chain Reaction (RT-PCR)

#### 4.6.1. Extraction of Total RNA from Cells

In this experiment, the RNeasy Mini kit (QIAGEN, Hilden, Germany) was utilized to inoculate 5 × 10^5^ cells in a 10 cm culture dish with FE25, FE25-lenti-ctrl, FE25-PRA, and FE25-PRB in separate plates. After 24 h of culture, the culture medium was removed, and the cells were washed twice with 1X PBS. Subsequently, the cells were treated with 0.05% trypsin to detach them from the dish, and the detached cells were collected. To lyse the cells and facilitate RNA extraction, 700 μL of RLT lysis buffer (Thermo Fisher, Waltham, MA, USA) was added. The solution was aspirated several times using a 1 mL micropipette until the cells were completely dissolved, resulting in a transparent solution. Following the manufacturer’s instructions, reagents were added to extract RNA from the lysed cells. Once the RNA extraction was complete, the RNA was rapidly cooled to −80 °C and the concentration of RNA was quantified.

#### 4.6.2. Preparation of cDNA

The cDNA preparation used Reverse-iT^TM^ 1st strand synthesis kit (ABgene, Advanciotechnologies Ltd., Epsom, UK). Approximately 1 μg total RNA and 1.0 μL anchored oligo dT, supplemented with diethylpyrocarbonate-treated water to a total volume of 12 μL, were incubated at 70 °C for 5 min and then placed on ice. Approximately 8.0 μL reaction mixture (4.0 μL 5× First-strand synthesis buffer, 2.0 μL dNTP mix 5 mM each, 1.0 μL 100 mM DTT, and Reverse-iT^TM^ RTase blend 1.0 μL) was added to the above mix followed by incubation at 47 °C for 50 min. Then the reaction was terminated at 75 °C for 10 min, and cDNA was stored at −20 °C for later use.

#### 4.6.3. qPCR

We used the ABI Step One Plus system (Applied Biosystems, Waltham, MA, USA) and FastStart Universal SYBR Green Master (ROX, Basel, Switzerland) gene expression analysis reagents to quantify gene expression. Glyceraldehyde-3-phosphate dehydrogenase (*GAPDH*) was used as an internal control to analyze gene expression levels.

In the quantification of *PRA* and *PRB* expression, the data were normalized to *GAPDH*. The relative quantification was compared with FE25 for fold-change analysis. The primer sequences are listed in Table 1.

### 4.7. Terminal Deoxynucleotidyl Transferase dUTP Nick End Labeling (TUNEL) Assay

Apoptosis was assayed using a TUNEL Assay Kit (Roche, Basel, Switzerland) following the manufacturer’s instructions. FE25, FE25-lenti-ctrl, FE25-PRA, and FE25-PRB cells were seeded at a density of 1 × 10^5^ cells per well in 12-well culture plates. Cultured cells were allowed to attach for 24 h. We then treated the cells with P4 (100 nM) for 48 h. Adherent cells were fixed in 4% paraformaldehyde. The TUNEL probe detected breaks in the DNA strands. The cells were then incubated in a permeabilization solution for 2 min on ice. The cells were washed twice with PBS, and the TUNEL reaction mixture was added, followed by incubation at 37 °C for 60 min in a humidified atmosphere in the dark. The samples were washed twice with PBS and observed under a fluorescence microscope.

### 4.8. Sensitivity to Chemotherapy Drugs

Carboplatin and paclitaxel are the first-line chemotherapeutic drugs for OC [43]. The effects of carboplatin (SINPHAR Pharmaceutical Co. Ltd., Yilan, Taiwan) and paclitaxel (Formoxol; Yung Shin Pharm. Ind. Co. Ltd., Taichung, Taiwan) on FE25, FE25-lenti-ctrl, FE25-PRA, and FE25-PRB cells were evaluated. 2500–3000 cells were placed in each well of a 96-well plate and cultured overnight. Cells were treated with various concentrations of chemotherapy drugs for 48 h. Cell viability was evaluated using an XTT solution. After adding XTT solution for 2–5 h at 38 °C, a spectrophotometer (DYNEX MRX II ELISA reader, Bustehrad, Czech Republic) at 450 nm wavelength and a reference wavelength of 650 nm was used for determining the cell number. The IC50 value was calculated using a nonlinear regression model in GraphPad Prism (version 9.0; GraphPad Software, San Diego, CA, USA).

### 4.9. Western Blot

For FE25, FE25-lenti-ctrl, FE25-PRA, and FE25-PRB cells, 1 × 10^6^ cells were placed in a 10 cm culture dish. Cultured cells were allowed to attach for 24 h. We then treated the cells with P4 (100 μM) for 48 h. We used a lysis buffer (150 mM NaCl, 50 mM Tris–HCl, pH 7.4, 1% Nonidet P-40) and a proteinase inhibitor cocktail (04693116001, Roche) to lyse the tested cells. We performed 10% sodium dodecyl sulfate-polyacrylamide gel electrophoresis and transferred the gel to a nitrocellulose membrane (Hybond-C Super; GE Healthcare, Little Chalfont, UK). The antibody employed was PRA/B (1:1000, #8757; Cell Signaling Technology, Danvers, MA, USA). Actin (#4970, Cell Signaling), GAPDH (#5014, Cell Signaling), and α-tubulin (#2144, Cell Signaling, 1:10,000) were used as internal controls. Membranes were then incubated with the antibodies described above. A secondary antibody (horseradish peroxidase-conjugated goat anti-mouse IgG; AS003, Abclonal, Woburn, MA, USA) was used to bind the primary antibody. Enhanced chemiluminescence reagents (ECL; GE Healthcare) were used to detect the bound antibodies.

Cell Signaling Technology’s products (Danvers, MA, USA), including the AKT (#9272, phospho-AKT, #9271) and extracellular signal-regulated kinase (ERK, #4695, phospho-ERK, #4270), were assessed for their role in cell growth signaling. Additionally, BCL2 (B cell lymphoma 2, #12789-1-AP, Proteintech, Chicago, IL, USA), XIAP (X-linked inhibitor of apoptosis, ab21278, Abcam, Cambridge, UK), and cleaved caspase-3 (#9664, Cell signaling) were utilized to examine apoptotic signaling. To determine protein quantification, the data were normalized to total protein loading (actin, GAPDH, or α-tubulin). Relative quantification was then compared to the original cells, enabling fold-change analysis.

### 4.10. Statistical Analyses

Data were presented as the mean ± standard deviation of at least three independent experiments. The Mann–Whitney U test was used to compare two independent variables, and one-way ANOVA with post hoc analysis and the Bonferroni test was used to compare three independent variables. Statistical analyses were performed using the GraphPad Prism 6 software (La Jolla, CA, USA). *p* < 0.05 was considered a significant difference.

## 5. Conclusions

This study suggests that PRA and PRB play distinct roles in regulating the behavior of FE25 cells (a pre-ovarian cancer cell line) and that targeting these receptors could be a potential therapeutic strategy for OC treatment. If PRA or PRB overexpression is observed in HGSC, progesterone could be considered as an adjuvant therapy for these specific cancer patients. Therefore, if PRA or PRB overexpression is observed in HGSC, the potential use of progesterone as an adjuvant therapy should be thoroughly discussed between the patient and their healthcare provider. However, further research is needed to confirm these findings and investigate the mechanisms underlying these effects.

## Figures and Tables

**Figure 1 ijms-24-11823-f001:**
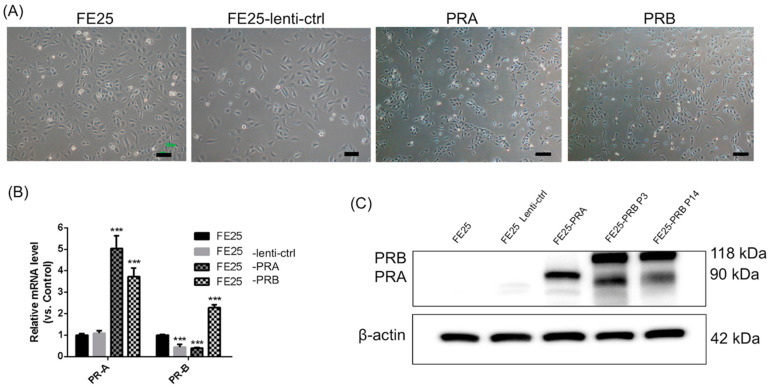
The morphology, mRNA, and protein expressions of FE25, FE25-lenti-ctrl, FE25-PRA, and FE25-PRB cells. (**A**) The morphology of various cells. The cell morphology was not changed after PRA or PRB transfection. The cell morphology revealed a cuboidal shape, which was characteristic of epithelial cells. Scale bar = 100 μm. (**B**) qPCR showed *PRA* and *PRB* expressions after transfection (n = 3). *** *p* < 0.001 when compared to FE25. FE25-PRA increased *PRA* expression. FE25-PRB increased both *PRA* and *PRB* expressions. (**C**) Western blot of PRA and PRB after *PRA* and *PRB* transfections. FE25-PRA increased PRA expression. The Western blot of PRB in FE25-PRB was performed at different passages (P3 and P14), which showed constant expression of PRB at different passages. PRA, progesterone receptor A; PRB progesterone receptor B; and qPCR, quantitative polymerase chain reaction.

**Figure 2 ijms-24-11823-f002:**
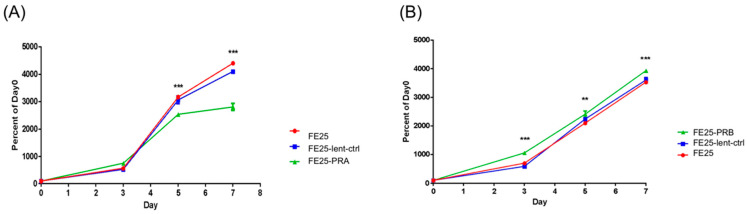
The proliferation of FE25-PRA, FE25-PRB, FE25, and FE25-lenti-ctrl. (**A**) FE25-PRA cells proliferated slower than FE25 and FE25-lenti-ctrl (n = 3 each). *** *p* < 0.001. (**B**) FE25-PRB cells proliferated faster than FE25 and FE25-lenti-ctrl (n = 3 each). ** *p* < 0.01, *** *p* < 0.01. All experiments were repeated in triplicate. PRA, progesterone receptor A; PRB, progesterone receptor B.

**Figure 3 ijms-24-11823-f003:**
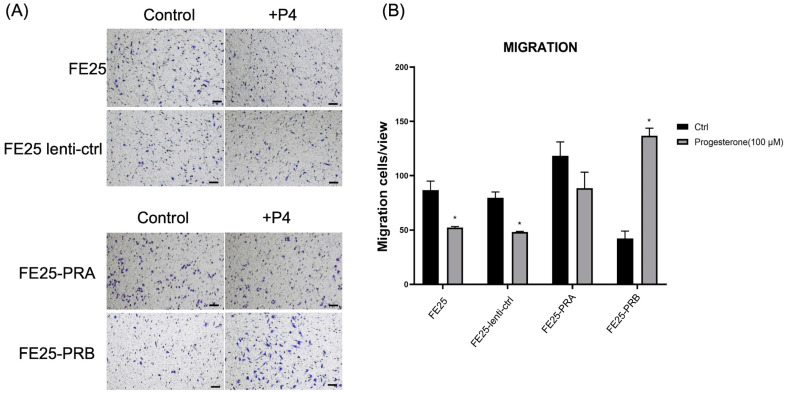
Migration assay of FE25, FE25-lenti-ctrl, FE25-PRA, and FE25-PRB with or without progesterone (100 μM) treatment. (**A**) Gross picture of migration cells stained with crystal violet. Scale bar = 100 μm. (**B**) Quantification of migrated cells. * *p* < 0.05 when compared to control (Ctrl). All experiments were repeated in triplicate. PRA, progesterone receptor A; PRB, progesterone receptor B; and P4, progesterone.

**Figure 4 ijms-24-11823-f004:**
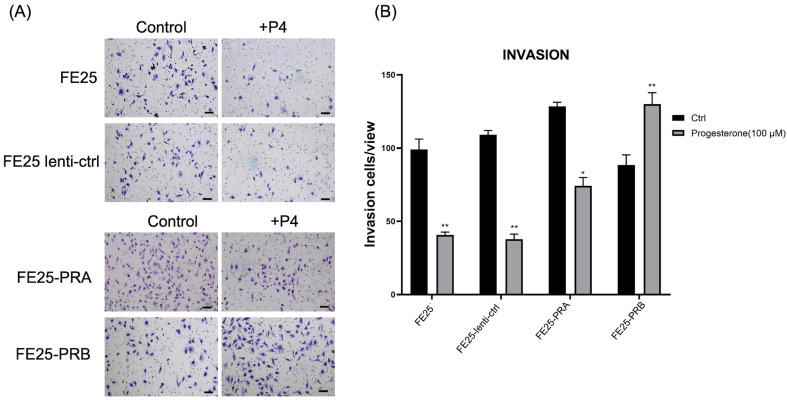
Invasion assay of FE25, FE25-lenti-ctr, FE25-PRA, and FE25-PRB with or without progesterone (100 μM) treatment. (**A**) Gross picture of invaded cells stained with crystal violet. (**B**) Quantification of migrated cells. * *p* < 0.05. ** *p* < 0.01 when compared to control (Ctrl). All experiments were repeated in triplicate. PRA, progesterone receptor A; PRB, progesterone receptor B; and P4, progesterone.

**Figure 5 ijms-24-11823-f005:**
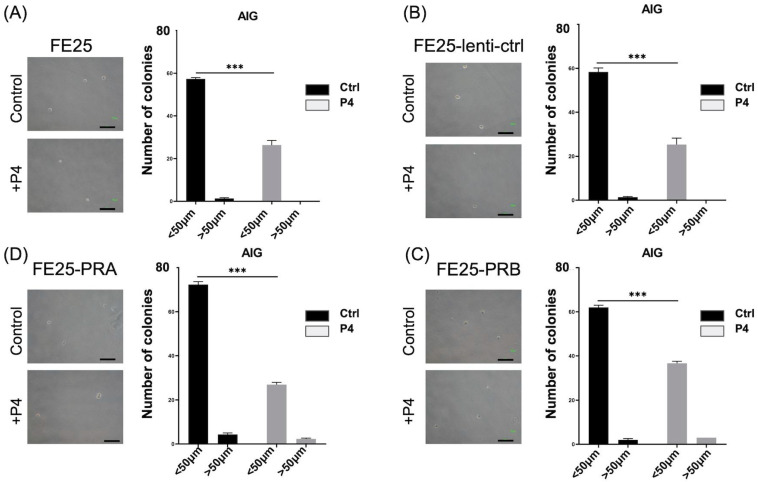
Anchorage-independent growth (AIG) in soft agar of various cells with or without progesterone (P4) 100 μM for 14 days. (**A**) FE25. (**B**) FE25-lenti-ctrl. (**C**) FE25-PRA. (**D**) FE25-PRB. Colony sizes < 50 μm and >50 μm were recorded under microscopy, respectively. In all groups, after P4 treatment, colony formation (<50 μm) was significantly decreased compared to that without P4 treatment. *** *p* < 0.001. Scale bar = 100 μm. P4: progesterone. PRA: progesterone receptor A. PRB: progesterone receptor B.

**Figure 6 ijms-24-11823-f006:**
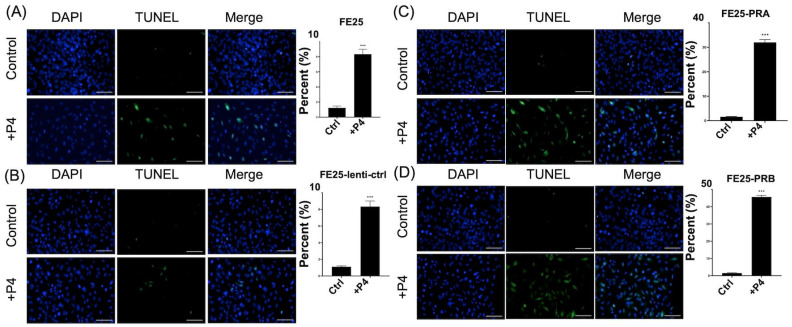
TUNEL assay for apoptosis of cells with or without progesterone receptor transfection after adding progesterone 100 μM. (**A**) FE25, scale bar = 100 μm. (**B**) F25-vector, scale bar = 100 μm. (**C**) FE25-PRA, scale bar = 100 μm. (**D**) FE25-PRB, scale bar = 100 μm. *** *p* < 0.001 compared to control (ctrl). Progesterone treatment increased TUNEL^+^ cells in all 4 cell lines. TUNEL, terminal deoxynucleotidyl transferase dUTP nick end labeling; PRA, progesterone receptor A; and PRB: progesterone receptor B. P4: progesterone.

**Figure 7 ijms-24-11823-f007:**
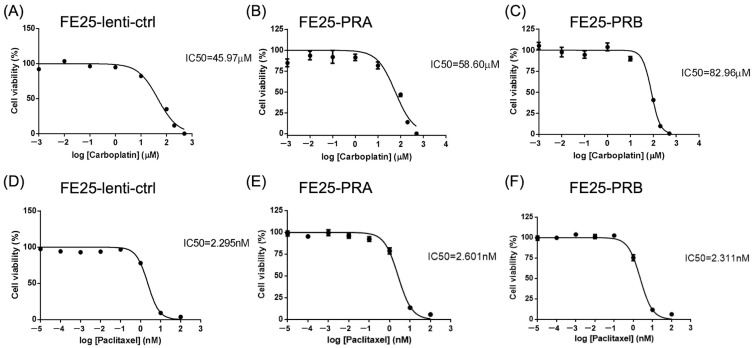
The half maximal inhibitory concentration (IC50) of carboplatin and paclitaxel on various cells. The IC50 values of carboplatin for FE25-vector (**A**), FE25-PRA (**B**), and FE25-PRB (**C**) after 24 h of chemotherapy. FE25-PRA or -PRB increased IC50 of carboplatin. The IC50 values of paclitaxel for FE25-vector (**D**), FE25-PRA (**E**), and FE25-PRB (**F**) after 24 h of chemotherapy. The IC50 of paclitaxel remained unchanged after PRA or PRB overexpression. PRA, progesterone receptor A; PRB, progesterone receptor B.

**Figure 8 ijms-24-11823-f008:**
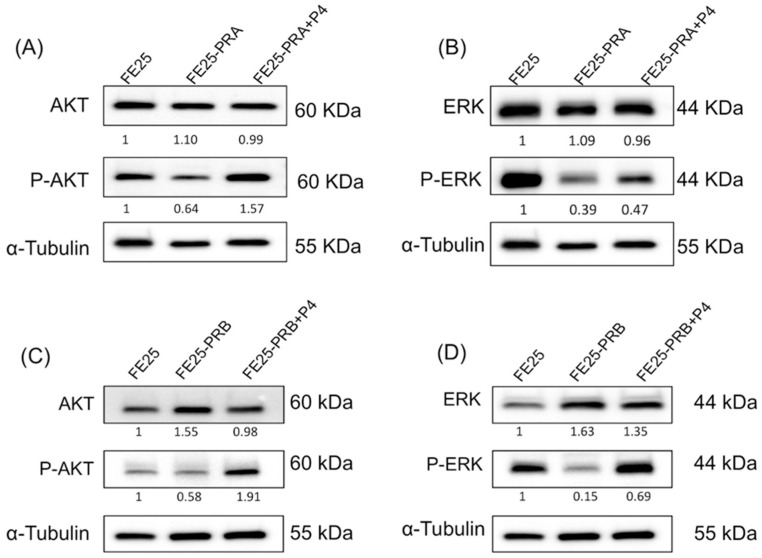
The AKT and ERK signaling pathways in FE25, FE25-PRA, and FE25-PRB without or with progesterone treatment. (**A**) Western blot of AKT expression in FE25, FE25-PRA +/− P4. (**B**) Western blot of ERK expression in FE25, FE25-PRA +/− P4. (**C**) Western blot of AKT expression in FE25, FE25-PRB +/− P4. (**D**) Western blot of ERK expression in FE25, FE25-PRB +/− P4. Protein levels of phospho-AKT (p-AKT) and phospho-ERK (p-ERK) were increased after progesterone treatment on FE25-PRA and FE25-PRB compared to FE25-PRA or FE25-PRB. After PRA or PRB-transfection, p-AKT and p-ERK were reduced relative to FE25. P4: progesterone, PRA: progesterone receptor A, PRB: progesterone receptor B; and ERK, extracellular signal-regulated kinase. The image was representative of n= 3. The number under each lane represented the relative quantitation of the expression in FE25.

**Figure 9 ijms-24-11823-f009:**
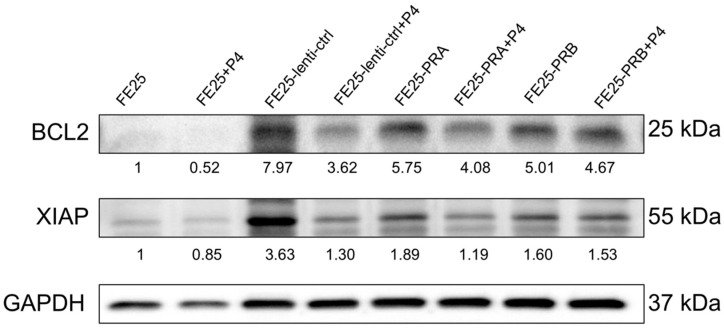
Progesterone activated BCL2 and XIAP signaling pathways. After adding progesterone, protein levels of BCL2 and XIAP were decreased. FE25, FE25-lenti-ctrl, FE25-PRA, and FE25-PRB were treated with progesterone 100 μM for 24 h, and a western blot was performed. BCL2, B cell lymphoma 2; XIAP, X-linked inhibitor of apoptosis; PRA: progesterone receptor A; PRB: progesterone receptor B. Representative of n=3. The number under each lane represented the relative quantitation of the expression in FE25.

**Figure 10 ijms-24-11823-f010:**
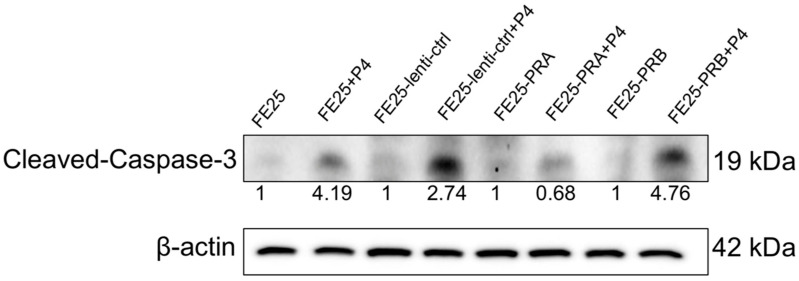
Progesterone activated the caspase signaling pathway. These cell lines were treated with progesterone 100 μM for 24 h, and a Western blot was performed. After adding progesterone, protein levels of cleaved caspase 3 were increased in FE25, FE25-lenti-ctrl, and FE25-PRB, but not FE25-PRA PRA, progesterone receptor A; PRB, progesterone receptor B. Representative of n= 3. The number under each lane represented the relative quantitation of the expression in FE25.

**Table 1 ijms-24-11823-t001:** Primer Sequences.

Gene	Forward 5′→3′	Reverse 5′→3′	Size (bp)
*PRB*	TATCTCCCTGGACGGGCTAC	TGTCCAAGACACTGTCCAGC	194
*PRA*	CGCGCTCTACCCTGCACTC	TGAATCCGGCCTCAGGTAGTT	121
*GAPDH*	GGTCTCCTCTGACTTGAACA	GTGAGGGTCTCTCTCTTCCT	221

Abbreviations: *PRA*, progesterone receptor A; *PRB*, progesterone receptor B; *GAPDH*; glyceraldehyde-3-phosphate dehydrogenase.

## Data Availability

All data are presented in the manuscript.

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
