# Peer review of "Effects of the Overexpression of Progesterone Receptors on a Precancer p53 and Rb-Defective Human Fallopian Tube Epithelial Cell Line"

_ijms, 2023, doi:10.3390/ijms241411823_

Round 1
Reviewer 1 Report
Abbreviations should be explained only in the first usage. Abbreviations should be checked.
There are some interpretation of the findings in the results section. They should be removed to the discussion section.
References have missing data (some page/article numbers are missing). They should be checked.
Author Response
Reviewer #1
Comment 1: Abbreviations should be explained only in the first usage. Abbreviations should be checked.
Response: In response to the comment, we have expelled the abbreviations only when they appeared the first time. The corrections can be found in Lines 23 (FE25-PRA, FE25-PRB), 90 (HGSC), 95 (AIG, IC50), 96 (P4), 99 (FE25), 106 (CMV), 117 (XTT), and 215 (BCL2), 216 (XIAP), and 356 (p-AKT, p-ERK).
Comment 2: There are some interpretation of the findings in the results section. They should be removed to the discussion section.
Response: We thank the reviewer for the suggestion, which is much appreciated. However, a comment from the reviewer #2 asked us to perform the opposite revision, i.e., move the interpretations in the discussion section to the result section. We kind of got stuck in the middle of responding to both suggestions and decided to leave these interpretations in the result section. We sincerely hope that the reviewer could understand our obstacle and approve our decision.
Comment 3: References have missing data (some page/article numbers are missing). They should be checked.
Response: We thank the reviewer for the suggestion. We have checked all the references and added the page numbers (Reference# 2, 9, and 30).
Reviewer 2 Report
In this study the authors aimed to investigate the impact of progesterone receptors A (PRA) and B (PRB) expression in a human fallopian tube epithelial cell line (FE25) with precancerous characteristics that does not express progesterone receptors. This cell line was previously developed and characterized by the authors. In this study they used lentiviral vectors to express either PRA or PRB and evaluated the effect of this overexpression on various cellular processes including proliferation, migration, invasion, anchorage-independent growth (AIG) and apoptosis both in the absence and in the presence of progesterone. They also evaluated the sensitivity of cells overexpressing PRA or PRB to chemotherapy drugs.
This is an interesting and well-executed study that contributes to expand the knowledge on an important topic in the field of ovarian cancer.
General comments:
Presentation of the results could be improved by better contextualizing each set of experiments. This chapter would also benefit from interpretation of the results presented. Some interpretations given in the discussion could be moved to the results chapter leaving in the discussion only the interpretation related to the integration of the results. Titles of the results chapters should also be revised for clarity and consistency. Figure legends could also be improved.
Specific comments:
Line 41: Instead of “ … been used to treat OC therapy.” should be “ … been used to treat OC.”
Lines 56-57: the sentence “In a previous study, the tumors were PR-negative and associated with poor survival.” should be “In a previous study, the tumors thar were PR-negative associated with poor survival.”
Lines 57-58: “…recruited 2933 OC patients with OC showed…” should be “…recruited 2933 patients with OC showed…”
Line 117-126: the procedure of extraction of total RNA from cells should be revised. It is confusing.
Figure 5 and 6: graphs are too small, and the axis titles are not legible.
Line 206: “PRA-transfected FE25 cells were increased PRA expression but not PRB than FE25 cells”. This sentence and the whole paragraph bellow is very confusing.
Line 316: “…phospho-AKT and phospho-ERK were decreased than those in FE25.” should be “…phospho-AKT and phospho-ERK were reduced relative to FE25.”
Overall quality of English is good but recommend revising all the manuscript for grammar errors.
Author Response
Reviewer #2
In this study the authors aimed to investigate the impact of progesterone receptors A (PRA) and B (PRB) expression in a human fallopian tube epithelial cell line (FE25) with precancerous characteristics that does not express progesterone receptors. This cell line was previously developed and characterized by the authors. In this study they used lentiviral vectors to express either PRA or PRB and evaluated the effect of this overexpression on various cellular processes including proliferation, migration, invasion, anchorage-independent growth (AIG) and apoptosis both in the absence and in the presence of progesterone. They also evaluated the sensitivity of cells overexpressing PRA or PRB to chemotherapy drugs.
This is an interesting and well-executed study that contributes to expand the knowledge on an important topic in the field of ovarian cancer.
General comments:
Presentation of the results could be improved by better contextualizing each set of experiments. This chapter would also benefit from interpretation of the results presented. Some interpretations given in the discussion could be moved to the results chapter leaving in the discussion only the interpretation related to the integration of the results. Titles of the results chapters should also be revised for clarity and consistency. Figure legends could also be improved.
Response: We thank the reviewer for the suggestion. We revised them accordingly.
For the first suggestion, we have improved the presentation of our results by better contextualizing each set of experiments (Please see the result section).
For the second suggestion, we have moved some interpretations given in the discussion section to the result section. These statements can be found in Lines 369-371 and 387-392.
For example, the statements in Lines 369-371 read as: “These results suggest that the presence of PRB may influence the response to progesterone treatment in terms of BCL2 and XIAP expression.” The statements in Lines 387-392 read as: “Progesterone was found to increase cleaved caspase-3 expression in FE25, FE25-lenti-ctrl, and FE25-PRB cells, indicating its pro-apoptotic effect in these cell types. However, in FE25-PRA cells, progesterone had the opposite effect, suggesting a distinct response in terms of cleaved caspase-3 expression. Overall, these findings highlight the complex and context-dependent effects of progesterone on cellular pathways and apoptotic processes in different cell lines.”
For the third suggestion, the sub-titles of the paragraphs in the result section have all been modified to improve the clarity and consistency (Please see the result section). In other words, we used a sentence to summarize the major findings of each set of experiments described in the paragraph. For example, we used “FE25-PRA Promoted Cell Migration, While FE25-PRB Inhibited Cell Migration, Both of Which Were Reversed by Progesterone Treatment” as the sub-title for the sub-section of 3.3 paragraph (Line 272).
For the fourth suggestion, we have improved the figure legends by giving additional information for the comprehension of the figures.
Specific comments:
Comment 1: Line 41: Instead of “ … been used to treat OC therapy.” should be “ … been used to treat OC.”
Response: In response to the comment, the sentence has been rewritten (Lines 46-48), and it reads as: “poly ADP ribose polymerase (PARP) inhibitors have emerged as treatment options for OC”
Comment 2: Lines 56-57: the sentence “In a previous study, the tumors were PR-negative and associated with poor survival.” should be “In a previous study, the tumors thar were PR-negative associated with poor survival.”
Response: In response to the comment, we have added “that” (Lines 77-78).
“In a previous study, the tumors that were PR-negative associated with poor survival [16].”
Comment 3: Lines 57-58: “…recruited 2933 OC patients with OC showed…” should be “…recruited 2933 patients with OC showed…”
Response: In response to the comment, we have revised it accordingly (Lines 78-80).
“recruited 2933 patients with OC showed that PR expression was associated with favorable survival in patients with endometrioid and high-grade serous carcinoma [17].”
Comment 4: Line 117-126: the procedure of extraction of total RNA from cells should be revised. It is confusing.
Response: In response to the comment, we have revised it to improve the clarity.
The revised paragraph (Lines 145-153) was “In this experiment, the RNeasy Mini kit (QIAGEN, Hilden, Germany) was utilized to inoculate 5×105 cells in a 10-cm culture dish with FE25, FE25-lenti-ctrl, FE25-PRA, and FE25-PRB in separate plates. After 24 hours of culture, the culture medium was removed, and the cells were washed twice with 1X PBS. Subsequently, the cells were treated with 0.05% trypsin to detach them from the dish, and the detached cells were collected. To lyse the cells and facilitate RNA extraction, 700 μl of RLT lysis buffer (Thermo Fisher) was added. The solution was aspirated several times using a 1 ml micropipette until the cells were completely dissolved, resulting in a transparent solution. Following the manufacturer's instructions, reagents were added to extract RNA from the lysed cells. Once the RNA extraction was complete, the RNA was rapidly cooled to -80℃ and the concentration of RNA was quantified.”
Comment 5: Figure 5 and 6: graphs are too small, and the axis titles are not legible.
Response: In response to the comment, the graphs have been enlarged and the sizes of the characters in the axis have also been magnified (Figures 5 and 6).
Comment 6: Line 206: “PRA-transfected FE25 cells were increased PRA expression but not PRB than FE25 cells”. This sentence and the whole paragraph bellow is very confusing.
Response: In response to the comment, the paragraph has been rewritten (Lines 234-245), and it reads as: “PRA-transfected FE25 cells exhibited increased expression of PRA, while PRB expression remained unchanged compared to FE25 cells (Figure 1B). To validate these findings at the protein level, we performed Western blotting to assess the levels of PRA and PRB proteins following transfection. Overexpressed PRA was noted in PRA-transfected FE25 cells (Figure 1C). Overexpression of both PRA and PRB proteins in PRB-transfected FE25 cells compared to untransfected or control vector-transfected FE25 cells (Figure 1C). Notably, overexpression of PRB led to an increase in both PRB mRNA and protein expression, as illustrated in Figure 1B and C. Additionally, we examined PRB expression at passages 3 and 14 and found consistent and stable expression levels (Figure 1C). Collectively, the morphology of FE25 cells remained unaltered following PRA or PRB overexpression. PRA overexpression resulted in increased PRA mRNA and protein expression, while PRB overexpression led to increased PRB mRNA and protein expression.”
Comment 7: Line 316: “…phospho-AKT and phospho-ERK were decreased than those in FE25.” should be “…phospho-AKT and phospho-ERK were reduced relative to FE25.”
Response: In response to the comment, we revised it accordingly (Line 358) and it reads as: “p-AKT and p-ERK were reduced relative to FE25.”
Comment 8: Comments on the Quality of English Language
Overall quality of English is good but recommend revising all the manuscript for grammar errors.
Response: In response to the comment, we have proofread the whole manuscript and revised the grammar errors throughout the text.
Reviewer 3 Report
This study evaluated the role of progesterone receptors on proliferation, invasion and migration of fallopian tube epithelial cell lines with defective tumour suppressor genes. I have the following comments:
1. In the introduction section the authors should further explain the subtypes of EOC and its precursor lesions
2. I do not understand the following statement: ‘Recently, targeted therapy with anti-vascular endothelial growth factor and poly ADP ribose polymerase inhibitors has been used to treat OC therapy’
3. What are the clinical response rates of various types of targeted therapies in OC?
4. What are the possible clinical implications of this study?
5. What are the response rates of antiprogesterone therapy in the treatment of EOC? Have there been any studies? What is the purpose of this study taking into consideration low response rates of various types of targeted therapies in OC?
Author Response
Reviewer #3
This study evaluated the role of progesterone receptors on proliferation, invasion and migration of fallopian tube epithelial cell lines with defective tumour suppressor genes. I have the following comments:
Comment 1: In the introduction section the authors should further explain the subtypes of EOC and its precursor lesions
Response: We thank the reviewer for this excellent suggestion. We have added a paragraph in the introduction section (5th paragraph; Lines 84-88) to explain the EOC subtypes and their precursor lesions.
These statements read as: “OC is divided into several subtypes: serous, mucinous, endometrioid, clear cell, Brenner tumor, and undifferentiated carcinoma [20]. The precursor lesion of serous carcinoma is the fallopian tube epithelium (FTE). Mucinous carcinoma may be originated from germ cells. Endometrioid and clear cell carcinoma may be originated from endometrial tissue. Brenner tumors may be originated from transitional cells [20].”
Comment 2: I do not understand the following statement: ‘Recently, targeted therapy with anti-vascular endothelial growth factor and poly ADP ribose polymerase inhibitors has been used to treat OC therapy’
Response: In response to the comment, we have revised the sentence to improve the clarity (Lines 46-49).
These statements read as: “Besides the above therapy, targeted therapies utilizing anti-vascular endothelial growth factor (VEGF) agents and poly ADP ribose polymerase (PARP) inhibitors have emerged as treatment options for OC [4]. These therapies have shown promising results in improving patient outcomes and are being incorporated into OC treatment strategies [4].”
Comment 3: What are the clinical response rates of various types of targeted therapies in OC?
Response: We thank the reviewer for allowing us to elaborate on this issue, which has been addressed in the introduction (1st paragraph; Lines 49-52).
These statements read as: “Response rates with bevacizumab-containing regimens range from approximately 30% to 60% in OC [5,6]. In clinical trials, PARP inhibitors have shown response rates ranging from approximately 40% to 60% in patients with BRCA-mutated OC [7,8].”
Comment 4: What are the possible clinical implications of this study?
Response: We thank the reviewer for allowing us to elaborate on this issue, which has been addressed in the conclusion section (Lines 498-501) as well as in the abstract section (Lines 34-36).
The statements in the conclusion read as: “If PRA or PRB overexpression is observed in HGSC, progesterone could be considered as an adjuvant therapy for these specific cancer patients. Therefore, if PRA or PRB overexpression is observed in HGSC, the potential use of progesterone as an adjuvant therapy should be thoroughly discussed between the patient and their healthcare provider.”
The statements in the abstract read as: “If PRA or PRB overexpression is observed in high-grade serous carcinoma, progesterone could be considered as an adjuvant therapy for these specific cancer patients.
Comment 5: What are the response rates of antiprogesterone therapy in the treatment of EOC? Have there been any studies? What is the purpose of this study taking into consideration low response rates of various types of targeted therapies in OC?
Response: We thank the reviewer for allowing us to elaborate on this issue, which has been addressed in the introduction section (2nd paragraph; Lines 54-65). These statements read as: “Previous studies have reported a variable response rate of antiprogesterone therapy, ranging from 5 % to 34 % [10]. In a retrospective cohort study, it was found that strong expression of progesterone receptor B (PRB) was associated with improved platinum sensitivity and overall survival [9]. Additionally, there was a suggestion that progesterone might enhance the effectiveness of platinum-based chemotherapy, indicating its potential as a platinum sensitizer [9]. Hence, PR status not only serves as a prognostic factor but also as a marker for chemotherapy efficacy in OC. Moreover, PRs have been shown to have distinct roles in regulating the behavior of ovarian cancer. Understanding these roles is crucial in comprehending the mechanisms underlying the disease. Furthermore, it is important to consider the economic burden associated with targeted therapies, which tends to be significantly higher compared to hormone therapies. Consequently, exploring the involvement of PRs in OC holds considerable significance in terms of both clinical outcomes and healthcare costs.”